# A Probiotic Formula for Modulation of Colorectal Cancer Risk via Reducing CRC-Associated Bacteria

**DOI:** 10.3390/cells12091244

**Published:** 2023-04-25

**Authors:** Jessie Qiaoyi Liang, Yao Zeng, Effie Yin Tung Lau, Yuting Sun, Yao Huang, Tingyu Zhou, Zhilu Xu, Jun Yu, Siew Chien Ng, Francis Ka Leung Chan

**Affiliations:** 1Department of Medicine and Therapeutics, Institute of Digestive Disease, Li Ka Shing Institute of Health Sciences, CUHK Shenzhen Research Institute, The Chinese University of Hong Kong, Hong Kong, China; 2Center for Gut Microbiota Research, Faculty of Medicine, The Chinese University of Hong Kong, Hong Kong, China; 3Department of Microbiology, The Chinese University of Hong Kong, Hong Kong, China; 4Microbiota I-Center (MagIC), Hong Kong, China

**Keywords:** colorectal cancer, probiotics, *Fusobacterium nucleatum*, *Bifidobacterium*

## Abstract

Gut microbiota dysbiosis with increased pathogenic bacteria and decreased beneficial bacteria is associated with colorectal cancer (CRC) development. This study examined the effect of a newly developed probiotic formula in modulating CRC-related bacteria. We developed a probiotic formula containing three bifidobacteria (*B. adolescentis*, *B. longum*, and *B. bifidum*) based on the identification of bacterial species that showed significant correlations with CRC-related bacteria including *Fusobacterium nucleatum* (*Fn*), *Lachnoclostridium sp. m3*, *Clostridium hathewayi* (*Ch*), and *Bacteroides clarus* (*Bc*). We co-cultured *Fn* with each bifidobacterium or the combined formula and examined the growth of *Fn* by qPCR. The three individual bifidobacteria significantly inhibited the growth of *Fn* compared to the control treatment (24~65% inhibition; all *p* < 0.001). The combination of the three bifidobacteria showed a greater inhibitory effect on *Fn* growth (70% inhibition) than the individual bifidobacteria (all *p* < 0.05). We further examined the effect of the probiotic formula in a pilot study of 72 subjects (40 on probiotics; 32 with no intervention) for 4 weeks and followed them up for 12 weeks. The relative fecal abundances of the bifidobacteria in the formula and the CRC-related markers (*Fn*, *m3*, *Ch*, and *Bc*) were quantitated by qPCR before and after the intervention, and the combined CRC risk score (4Bac; *Fn, m3*, *Ch*, and *Bc*) was evaluated. Subjects with probiotics intervention showed significantly increased abundances of the bifidobacteria from week 2 to week 5 compared to baseline (*p* < 0.05), and the abundances dropped to baseline levels after the cessation of the intervention. There were significant decreases in the levels of CRC-related markers (*Fn* and *m3*) and the CRC risk score (4Bac) from week 2 to week 12 compared to baseline levels (*p* < 0.05) in the intervention group but not in the control group. A novel probiotic formula containing *B. adolescentis*, *B. longum*, and *B. bifidum* was effective in inhibiting the growth of *F. nucleatum* in vitro and improving the gut microbial environment against CRC development.

## 1. Introduction

Gut microbiota dysbiosis has been associated with colorectal tumorigenesis [1,2,3,4]. Specifically, some pathogenic bacteria, such as *Fusobacterium nucleatum*, have been shown to play important roles in CRC development [5,6]. Therefore, it is anticipated that reducing or eliminating pathogenic bacteria associated with CRC can reduce the risk of CRC development. Probiotics can exert beneficial effects on gut microbiota and help maintain homeostasis [7]. Probiotics can directly influence colonization of microbes via the production of inhibitory compounds (bacteriocins, short-chain fatty acids, etc.) and substrates that might nourish other microbes (secreted exopolysaccharides, vitamins, etc.). Probiotics can also indirectly modulate microbiota by affecting the host immune system and intestinal barrier integrity [7]. The anti-tumor effects of probiotics against CRC have been reported. In vitro studies have demonstrated that the co-culturing of certain probiotic strains inhibit the proliferation of and induce apoptosis in colon cancer cells [8,9]. In vivo studies have shown the effectiveness of probiotics in reducing cancer incidence or suppressing tumor growth in carcinogen-treated animals [10,11]. In human studies, probiotics have been mainly used as an adjuvant treatment during chemotherapy, but fewer have been used in prevention due to study difficulties. They have also shown to be associated with a reduced risk of post-operative complications. An earlier prospective study from Italy showed that subjects with an increased consumption of yogurt for 12 years had lower CRC incidence, suggesting that microbiota modulation via diet may play a role in CRC prevention, although the mechanisms supporting this benefit have not been elucidated [12].

However, it is unknown whether probiotic bacteria inhibit CRC development via the amelioration of gut microbiota dysbiosis. Whether the pathogenic bacteria involved in CRC development can be inhibited by specific probiotic bacteria has not been investigated, and nor have their mechanisms of action. We hypothesize that bacteria associated with CRC development can be altered by microbiota modulation via a targeted probiotic formula. By using metagenomics analysis to compare the fecal microbiome of CRC patients and healthy subjects, we identified 20 bacterial gene markers for the non-invasive diagnosis of CRC, 8 of which were enriched in CRC patients (8Up), while the other 12 were decreased in CRC patients (12Down) [1]. Using targeted quantification by quantitative PCR (qPCR), we further demonstrated that a panel of four bacterial markers (4Bac), composed of *Fusobacterium nucleatum* (*Fn*), *Bacteroides clarus* (*Bc*), *Clostridium hathewayi* (*Ch*), and *Lachnoclostridium* sp. (*m3*), showed good diagnostic performance for colorectal adenoma (including in the non-advanced stage) and CRC [13]. These findings pave the way for assessing CRC risk based on bacterial markers using qPCR or metagenome sequencing. In this study, we first developed a probiotic formula based on the metagenome sequencing data of a CRC cohort to identify probiotic species that inversely correlate with our previously identified bacterial markers for CRC risk. We then investigated the effects of a formula containing the identified probiotic species in reducing pathogenic bacteria of CRC in human subjects.

## 2. Materials and Methods

### 2.1. Metagenomics Dataset

We analyzed fecal metagenomic sequencing data from 589 Hong Kong Chinese subjects (184 CRC, 185 adenoma, and 220 control subjects) consisting of a discovery cohort of 74 subjects with CRC and 54 controls for the identification of the 20 CRC-related bacterial gene markers [1,14]. This study has been approved by The Joint Chinese University of Hong Kong, New Territories East Cluster Clinical Research Ethics Committee (The Joint CUHK-NTEC CREC, CREC Ref. No: 2021.126). Written informed consent was obtained from all subjects. Abundances of the 20 gene markers were analyzed as described in our previous study [13]. Relative abundances of species were analyzed by MetaPhlAn3 [15]. 

### 2.2. Design of a Probiotic Formula against CRC-Associated Bacteria

Firstly, Spearman correlation was analyzed between abundances of all detected probiotics (22 species detected in our cohort as listed in Figure 1A) and disease development from normal to adenoma and further to CRC to identify probiotic species that significantly decreased with disease development. To identify probiotic species that potentially modulate CRC-related bacteria, we analyzed correlations in the abundances between the probiotic species and 20 CRC-related bacterial gene markers previously identified by our team [1]. The combined scores of the 8 markers increased in CRC (8Up) and the 12 markers decreased in CRC (12Down) were included in correlation analysis. We also included the four individual gene markers that were targeted by our qPCR test for non-invasive diagnosis of CRC and adenoma, mapping to *Fusobacterium nucleatum* (*Fn*), *Lachnoclostridium* sp. *m3*, *Clostridium hathewayi* (*Ch*), and *Bacteroides clarus* (*Bc*) [13]. Probiotic species inversely correlated with markers enriched in CRC or positively correlated with markers decreased in CRC were selected. Finally, three Bifidobacterium species that significantly decreased with CRC development and significantly correlated with CRC-related markers were included in the probiotic formula, including *B. adolescentis*, *B. longum*, and *B. bifidum*. For clinical trial, we also included three specific prebiotics (Galactooligosaccharides, xylooligosaccharide, and resistant dextrin) in the formula to stimulate the favorable growth and/or enhance activities of the probiotic bacteria [16,17,18,19]. Capsules containing the three bifidobacteria (25 billion CFU per capsule) and prebiotics were prepared as described in our previous study [20].

### 2.3. Bacterial Strains and Culture Conditions

The bifidobacterium strains (*B. adolescentis* DSM 18351, *B. longum* DSM 16603, and *B. bifidum* DSM 22892) were obtained from Probiotical (Novara, Italy). *Fn* (ATCC 25586) and *Enterocloster aldenensis* (ATCC BAA 1318) were obtained from American Type Culture Collection (Manassas, VA, USA). All the bacteria were cultured in RCM medium (Sigma-Aldrich, St. Louis, MO, USA) under anaerobic condition at 37 °C. 

### 2.4. Bacterial Co-Culture Assay

Co-culture experiments were performed to investigate whether the selected bifidobacteria inhibited the growth of *Fn*. *B. adolescentis*, *B. longum*, and *B. bifidum*, either individually or in combination, were co-cultured with *Fn*, with *E. aldenensis* (an anaerobic species that has shown no correlation with *Fn*) used as control treatment. Specifically, five co-culture groups in final volumes of 20 mL were analyzed: (1) *B. adolescentis* (10^9^ CFU) plus *Fn* (10^9^ CFU), (2) *B. longum* (10^9^ CFU) plus *Fn* (10^9^ CFU), (3) *B. bifidum* (10^9^ CFU) plus *Fn* (10^9^ CFU), (4) probiotics combination (10^9^ CFU) plus *Fn* (10^9^ CFU), and (5) *E. aldenensis* (10^9^ CFU) plus *Fn* (10^9^ CFU). The ratio of *B. adolescentis* to *B. longum* to *B. bifidum* is 4.6:2.7:2.7. Each treatment was repeated in triplicate. All treatment/control groups started with the same copies of *Fn* in the same volumes (20 mL) and were cultured under anaerobic conditions at 37 °C for 12 h. After that, the culture media containing bacteria were mixed fully, and 10 μL of each were diluted with 90 μL ultrapure water (total 100 μL) and boiled at 100 °C for 30 min for bacterial DNA extraction. Then, 2 μL of each of the 100 μL DNA extracts were used as templates in each qPCR reaction. A set of primers and a probe carrying a 5′ reporter dye FAM (6-carboxy fluorescein) specifically targeting *Fn*, as listed in Appendix A, was used. qPCR amplifications were performed in a 20 µL reaction system of TaqMan Universal Master Mix II (Applied Biosystems) containing 0.3 µM of each primer and 0.2 µM of the probe in MicroAmp fast optical 96-well reaction plates (Applied Biosystems, Waltham, MA, USA) with adhesive sealing. Thermal cycler parameters were 95 °C for 10 min and (95 °C 15 s, 60 °C 1 min) × 45 cycles on an ABI QuantStudio sequence detection system. The growth of *Fn* was then assessed as the relative abundance of *Fn* in each sample (equivalent to equal volumes of cultures) using ∆Cq method compared to the control group (Power (2, −(Cq_treatment/control_ − Cq_control_))) and shown as percentages in reference to control treatment.

### 2.5. Pilot Clinical Study

We have previously conducted a pilot clinical trial in which intervention using the probiotics formula containing *B. adolescentis*, *B. longum*, and *B. bifidum* was included [20]. Here, we made use of these samples to test the effects of this probiotics formula on modulating the abundances of CRC-related bacteria. This was an open-label study of consecutive hospitalized patients with COVID-19 in a tertiary referral center in Hong Kong. Patients were excluded if they were below the age of 18, on mechanical ventilation, admitted to intensive care unit, on peritoneal dialysis or haemodialysis, immunocompromised, had an active or known history of infective endocarditis, pregnant, or had a history of suspected intolerance to the probiotic formula or its components. The latter was defined as any condition such as allergic reaction or any discomfort that rendered the subject not suitable to participate in this study. A designated pharmacist was responsible for dispensing the study capsules. Forty patients (age: 50.4 ± 12.3y (mean ± SD); 24 males) received two doses of the probiotic formula (100 billion CFU) per day, while 32 patients (age: 50.3 ± 15.1y; 15 males) in the control group received no probiotic intervention. All the subjects were recruited consecutively, and the baseline stool samples were collected before probiotic administration. There were no significant differences in the age (*p* = 0.99) and sex (*p* = 0.24 by Fisher’s exact test) distributions between the two groups. All patients took study capsules together with standard meals for 4 weeks. Both intervention group and no-intervention group received the same treatment protocol for COVID-19 endorsed by the local health authority. This study was approved by the Clinical Research Ethics Committees (2020.407) and was registered in the Clinical Trials Registry (NCT04581018). Written informed consent was obtained from all patients. Stool samples were collected at baseline (week 0), week 2, week 4, week 5, week 8, and week 12 from all participants in both intervention group and no-intervention group. Stools were collected in Norgen’s Stool Preservative (Norgen Biotek Corp, Thorold, ON, Canada) and stored in a −80 °C freezer within 24 h until further analysis.

### 2.6. DNA Extraction and Quantification of Microbial Markers by Duplex qPCR

Fecal DNA was extracted using Maxwell RSC PureFood GMO and Authentication Kit (Promega, Madison, WI, USA) following manufacturer’s instructions. Fecal levels of four microbial DNA markers for CRC (*Fn*, *m3*, *Bc*, and *Ch*) and the three biofidobacteria (*B. adolescentis*, *B. longum*, and *B. bifidum*) were quantified by qPCR. Primer and probe sequences targeting the markers and 16s rDNA internal control are listed in Appendix A. Each probe carried a 5′ reporter dye FAM (6-carboxy fluorescein) or VIC (4,7,2′-trichloro-7′-phenyl-6-carboxyfluorescein) and a 3′ quencher dye TAMRA (6-carboxytetramethyl-rhodamine). Primers and hydrolysis probes were synthesized by Invitrogen (Carlsbad, CA, USA). qPCR amplifications were performed on an ABI QuantStudio sequence detection system as previously described, with thermal cycler parameters of 95 °C for 10 min and (95 °C 15 s, 60 °C 1 min) × 45 cycles [21]. Positive controls of the markers and a negative control (H2O as template) were included within every experiment. Measurements were performed in triplicates for each sample. Relative level of each marker was calculated using ∆Cq method, compared to internal control (Power (2, −(Cq_target_ − Cq_control_))), and shown as log value of ‘*10e6+1′.

### 2.7. Scoring Algorithms

Among the 20 CRC-associated gene markers, 8 were increased and 12 were decreased in CRC patients compared to control subjects. The 8Up score was calculated as Log_10_ [Sum(8 increased markers) +*1e-20]. The 12Down score was calculated as Log_10_ [Sum(12 decreased markers) +*1e-20]. The combined score of four microbial markers (4Bac) using a logistic regression model (4Bac score = 0.23162 × *Fn* + 0.13451 × *m3* − 0.10075 × *Bc* + 0.32841 × *Ch* − 2.73836) was determined in our previous study [13]. 

### 2.8. Statistical Analysis

Values were all expressed as mean ± SD or median (interquartile range (IQR)) as appropriate. The differences in bacterial levels between two groups were determined by Mann–Whitney U test or paired *t*-test. Correlations between bacterial species, abundances/gene marker, and abundances/scores were conducted by Pearson correlation analysis. Correlations between bacterial species abundances and colorectal neoplasm stage were analyzed by Spearman’s correlation analysis. The changes in markers across different time points were analyzed by one-way ANOVA multiple comparison for linear trend where appropriate. All tests were conducted by GraphPad Prism 9.4.1 (GraphPad Software Inc., San Diego, CA, USA) or MedCalc^®^ Statistical Software version 19.6 (MedCalc Software Ltd., Ostend, Belgium; https://www.medcalc.org; accessed on 20 December 2020). *p* < 0.05 was considered statistical significance.

## 3. Results

### 3.1. Identification of Bacterial Species Inversely Correlated with CRC Development from Human Metagenomic Datasets

Based on the Spearman correlation analysis of the probiotic species detected in our cohort (22 species detected), we identified three bifidobacterial species, including *B. adolescentis* (rho = −0.144; *p* < 0.001), *B. longum* (rho = −0.092; *p* < 0.05), and *B. bifidum* (rho = −0.081; *p* < 0.05), that inversely correlated with disease development from the control samples to the CRC samples (Figure 1A). On the other hand, other probiotics showed no significant trends of change, and *Lactobacillus salivarius* showed an increasing trend from the normal subjects to the CRC patients (*p* < 0.0001) (Figure 1A). Bacteria known to be associated with CRC development, such as *F. nucleatum* (*Fn*) and *Clostridium hathewayi* (*Ch*), were significantly increased in stool samples from the normal group to the CRC group, while beneficial species, such as *Roseburia intestinalis* and *Bacteroides clarus* (*Bc*), showed decreasing trends (Figure 1A). Importantly, the abundances of *B. adolescentis, B. longum*, and *B. bifidum* in stool were significantly decreased in subjects with CRC compared to those of the controls, and *B. adolescentis* and *B. longum* were also significantly decreased in subjects with adenoma compared to the controls (all *p* < 0.05; Figure 1B). 

### 3.2. Identification of Probiotic Species with Inverse Correlation with CRC-Associated Microbial Risk

The correlations with CRC markers also identified the same three bifidobacterial species (*B. adolescentis*, *B. longum*, and *B. bifidum*) as being significantly associated with one or more of the “pathogenic” CRC-related bacterial gene markers (*Fn*, *Ch*, *m3*, and 8Up) and of the “beneficial” markers (*Bc* and 12Down) (Figure 2A). *B. adolescentis* and *B. longum* were inversely correlated with *Fn* (Pearson’s r of −0.1 and −0.096, respectively, both *p* < 0.05). *B. longum* was positively correlated with *Bc* (r = 0.106, *p* = 0.01). *B. adolescentis*, *B. longum*, and *B. bifidum* were positively correlated with 12Down (Pearson’s r of 0.135, 0.134 and 0.126, respectively, all *p* ≤ 0.002) (Figure 2B). We further checked the fold changes of the CRC gene markers with the presence of the bifidobacteria compared to the samples without the corresponding bifidobacteria. The presence of each of the three bifidobacterial species was significantly associated with a decrease in at least one of the “pathogenic” markers (*Fn*, *Ch*, *m3*, and 8Up) and an increase in at least one of the “beneficial” markers (*Bc* and 12Down) (Figure 2C). These data support the potential effect of *B. adolescentis*, *B. longum*, and *B. bifidum* in CRC prevention based on the inverse correlation with CRC-associated microbial risk markers.

### 3.3. B. adolescentis, B. longum, and B. bifidum Suppressed the Growth of F. nucleatum In Vitro

We next evaluated whether *B. adolescentis*, *B. longum*, and *B. bifidum* can inhibit the growth of *Fn*. The co-culturing of the individual bifidobacteria and the combined three bifidobacteria with *Fn* led to an inhibition of the growth of *Fn* compared to the control treatment (26~70% inhibition; all *p* < 0.0001). The combination of the three bifidobacteria showed a significantly greater inhibitory effect on *Fn* growth (70% inhibition) than the individual bifidobacteria (*B. adolescentis* 65%, *B. longum* 26%, and *B. bifidum* 62%; all *p* < 0.05) (Figure 3), which is also greater than the sum of the proportions of the individual effects (53.7%). These data support the synergistic effect of the combination of three bifidobacterial species on suppressing the growth of *Fn*.

### 3.4. Intervention with B. adolescentis, B. longum, and B. bifidum Significantly Reduced Microbial Risk for CRC

We further evaluated the changes of the CRC-associated microbial markers (*Fn, m3, Ch*, and *Bc*, as well as their combined score 4Bac) in human subjects after taking a high dose of the probiotic formula, supplemented with prebiotics, for four weeks compared to the no-intervention control subjects (Figure 4A). At baseline, there was no significant difference in the relative abundances of the three *Bifidobacterium* species or CRC-markers (*Fn*, *m3*, and the combined 4Bac score) in fecal samples between the intervention and control groups (Figure 4B). In the intervention group, the relative abundances of *B. adolescentis*, *B. longum,* and *B. bifidum* significantly increased at week 2 and week 4/5 by a pairwise comparison to baseline levels (*p* < 0.05), and they decreased after stopping probiotics intake and showed no significant difference compared to baseline at weeks 8 and 12 (Figure 5A1). The abundances of *Fn* and *m3* and the 4Bac score were significantly decreased compared to baseline at almost all time points (except m3 at week 2) from week 2 to week 12 (2 months after stopping probiotics intake) (all *p* < 0.05 by paired *t*-test; Figure 5A1). By observing the trends of change, we found increasing trends in the three *Bifidobacterium* species from baseline to week 4, although only the increase in *B. bifidum* was significant (linear trend *p* < 0.05). The abundances of the three bifidobacteria dropped after stopping the probiotics intake, and the decrease in *B. bifidum* from week 4 to week 12 was significant (*p* < 0.05) (Figure 5A2). *Fn*, *m3,* and 4Bac dropped significantly from baseline to week 4 or from baseline to week 12 (linear trends all *p* < 0.05 except for *Fn* (week 0–12); Figure 5A2). In the no-intervention controls, no significant changes in the three bifidobacterial species or the CRC-markers were observed from week 2 to week 12 compared to baseline (Figure 5B1,B2).

## 4. Discussion

This is the first study providing evidence for a probiotic formula in reducing CRC-related microbial risk, thus implying the potential for CRC prevention. This probiotic formula, involving three bifidobacteria (*B. adolescentis*, *B. longum,* and *B. bifidum*), was established based on the identification of probiotic species that decreased significantly with colorectal neoplasia progression and was inversely correlated with CRC-enriched markers including *Fn*. We showed the effects of the selected bifidobacteria on suppressing the growth of *Fn* in vitro. Moreover, our clinical intervention trial further demonstrated the usefulness of the probiotic formula in reducing the microbial risk for CRC development as indicated by our previously devised test for CRC risk assessment. 

The anti-tumor effects of *B. adolescentis*, *B. longum,* and *B. bifidum* have been reported previously but only individually and mainly in in vitro and animal studies. Whole cells, cell extracts and cell-free supernatants of *B. adolescentis* have been shown to inhibit the proliferation of colon cancer cells in vitro [22,23]. *B. bifidum* has been shown to exert a cytotoxicity effect on colon cancer cells [24]. The administration of *B. bifidum* attenuated tumorigenesis in azoxymethane (AOM)/dextran sulphate sodium (DSS)-induced colitis-associated colon cancer in mice via modulating gut microbiota and metabolome [25]. *B. longum* has also been shown to reduce inflammation and tumor incidence in AOM/DSS-induced colon cancer in mice [26]. The combination of five bifidobacterium strains (1 *B. longum* strain, 2 *B. bifidum*, and 2 *B. breve* strains) has been shown to induce the apoptosis of colon cancer cells and significantly reduce the incidence and inhibit the progression of tumors in CRC mouse models [27]. However, these findings mainly focused on the effects of the probiotics on suppressing the growth of cancer cells in vitro or tumors in carcinogen-treated mice, which seems more for therapeutics.

On the other hand, here, we showed the effect of our probiotic formula on reducing levels of CRC-associated bacteria, which is indicative of the preventative potential against CRC development. Clinical results (Figure 1 and Figure 2) showed opposite trends of change from normal controls via adenoma to CRC or inverse correlations between the bifidobacteria and *Fn*, and *B. adolescentis* and *B. longum* were significantly decreased in adenoma patients compared to the control subjects. These data demonstrate that the bifidobacteria decreased at an early stage of CRC development, and their decrease may provide a more suitable intestinal environment for the growth of *Fn*. These data are in line with the in vitro (Figure 3) data, which showed the effects of the selected bifidobacteria on suppressing the growth of *Fn* in vitro. Abnormality in the composition of the gut microbiota has been implicated as a potentially important etiological factor in the initiation and progression of CRC [28]. With the widespread application of metagenomic analyses in the investigation of gut microbiota, an increasing number of bacteria have been identified to be positively associated with CRC [1,2,3,4]. Recent basic research has established a critical function for the dysbiotic microbiota [29] and specific bacterial species, such as *Fn* and *Ch* in promoting colorectal tumorigenesis. It has been well demonstrated that *Fn* induces inflammation and modulates the host immune response to promote tumor development [5,6]. *Ch* has been shown to promote colonic epithelial cell proliferation in mouse models [30]. The *Lachnoclostridium* sp. *m3* is a novel pathogenic species that potentially contributes to colorectal adenoma and cancer development [13]. *Bc* was found to be depleted in CRC patients in our previous study [21]. Although the role of *Bc* remains largely unexplored, some species of *Bacteroides* are considered to be the next generation of probiotics [31]. Therefore, the decreased levels of *Fn*, *Ch,* and *m3* and the increased level of *Bc*, contributing to a decreased combined score of 4Bac, reflect a decreased microbial risk for developing CRC. With the significantly reduced levels of 4Bac, the effect of our probiotic formula on reducing microbial risk for CRC was well demonstrated in this study. Probiotics suppress the growth of pathogens by the production of inhibitory compounds, such as bacteriocins and organic acids [32], and the competition of colonization sites with pathogens [33]. The mechanisms by which the three bifidobacteria suppressed the growth of *Fn*, as well as other CRC-associated bacteria, warrant further investigation.

The probiotic formula used in the clinical trial was supplemented with prebiotics, which may also contribute to the modulation of gut microbiota. However, as the dosage of prebiotics was much smaller than the daily intake of dietary fiber of the study subjects, its influence on the gut microbiota would have been limited. The bifidobacteria in the intervention group detected by qPCR at week 2 to 4/5 were probably the transient passages and the probiotics failed to colonize the colon. While most probiotics are transient in nature, many other factors influence the probiotic viability and mucoadhesive properties [34]. Therefore, strategies to improve the colonization of probiotics, or determining the appropriate time and dosage of probiotic supplementation, are important for future clinical application. Regardless, according to the data from our clinical trial, although the bifidobacteria cannot colonize in the subjects’ colons and resume to baseline level within one month of the discontinuance of probiotics intake, the effect on the reduced CRC markers can last for at least another eight weeks after the daily intake of the probiotic formula, supplemented with prebiotics, for four weeks. As the clinical trial involved patients who already had gut dysbiosis, the effectiveness of the probiotic formula on reducing CRC-associated bacteria and modulating CRC-associated microbiome function needs to be further verified or investigated by future clinical trials involving subjects representative of the general population.

Different strains of the same probiotic species may show differential properties, such as *B. adolescentis* of various origins [35]. *B. adolescentis* isolated from the feces of a new-born increased the body weight of mice, while those isolated from elderly humans significantly decreased body weight and increased serum leptin concentrations and the relative abundances of potentially beneficial genera (e.g., *Bacteroides*, *Parabacteroides*, and *Faecalibaculum*). The strains of the *B. adolescentis*, *B. longum,* and *B. bifidum* that were present in the subjects of our metagenome sequencing cohort might be different from the probiotic strains used in the in vitro experiments and the pilot clinical trial, which were originally isolated in non-Chinese populations. In addition to characterizing strain-specific functions, whether the strains isolated from a certain population would be more efficient at colonizing this population deserves further investigation.

Future studies are warranted to further verify the effects of the identified bifidobacterial species and the probiotic formula. In vitro assays are needed to test their inhibiting effects on the growth of other pathogenic bacteria for CRC. The suppressive effects and mechanisms of the identified bifidobacteria against the growth-promoting effects of *Fn*, *Ch,* and *m3* on colon cancer cells should be investigated. In vivo studies are also warranted to assess the formula in preventing/suppressing CRC development. Most importantly, a long-term prospective clinical study is warranted to demonstrate the CRC prevention effect of the formula.

In summary, a novel probiotic formula consisting of *B. adolescentis*, *B. longum,* and *B. bifidum* was effective in inhibiting the growth of *F. nucleatum* and improved the gut microbial environment against CRC development. The new probiotic formula composed of food grade components will be suitable to be taken by average risk adults to reduce microbial risk for CRC development. 

## Figures and Tables

**Figure 1 cells-12-01244-f001:**
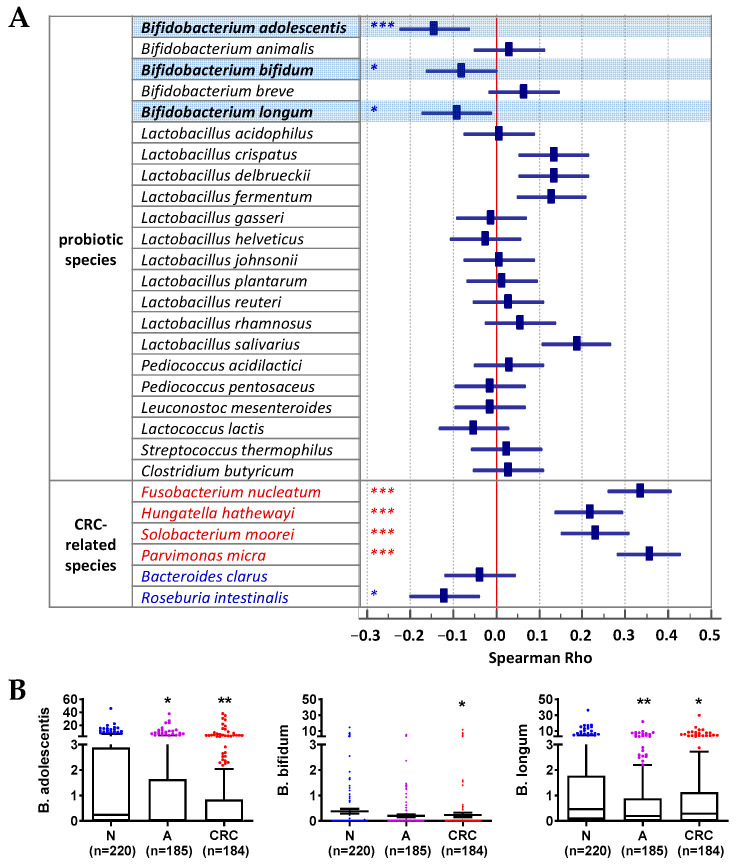
Correlations of probiotics with colorectal neoplasm development. (**A**) Spearman correlation of detected probiotic species and CRC-related species with colorectal neoplasm development. * *p* < 0.05, *** *p* < 0.0001; blue denotes decrease, and red denotes increase. (**B**) Relative abundances of species of interest in fecal samples from normal subjects (N), patients with adenoma (A), and CRC. * *p* < 0.05, ** *p* < 0.001 vs. N.

**Figure 2 cells-12-01244-f002:**
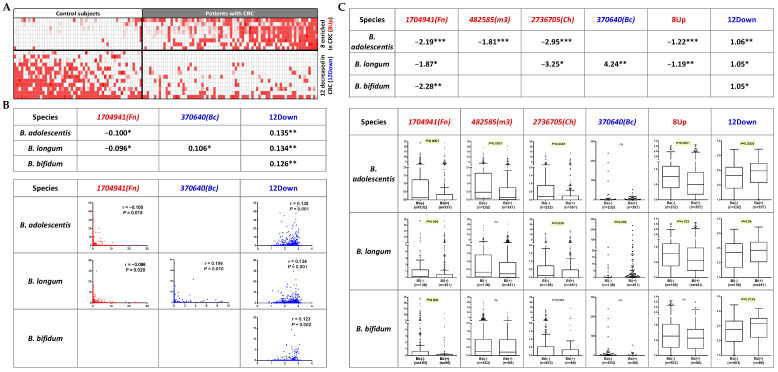
Correlations of probiotics species and CRC-related bacterial gene markers. (**A**) Heatmap illustration of the 20 bacterial gene markers identified by our previous study, with 8 significantly enriched and 12 significantly decreased in CRC patients compared to in healthy subjects. White color denotes relative abundance of 0. (**B**) Three bifidobacterial species (*B. adolescentis*, *B. longum*, and *B. bifidum*) inversely correlate with the pathogenic markers (shown in red) or positively correlate with the beneficial markers (shown in blue). (**C**) Fold changes of the CRC-associated gene makers in samples negative in *B. adolescentis*, *B. longum*, or *B. bifidum* compared to those positive in the corresponding bifidobacteria. * *p* < 0.05; ** *p* < 0.001; *** *p* < 0.0001; ns, not significant.

**Figure 3 cells-12-01244-f003:**
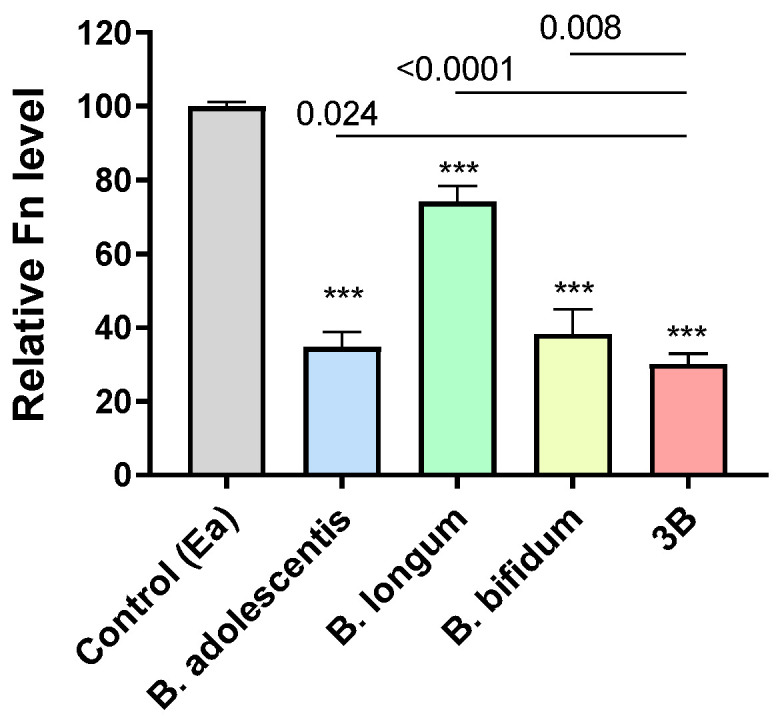
In vitro co-culturing with bifidobacteria significantly inhibited the growth of *F. nucleatum* (*Fn*) compared to the control treatment by *E. aldenensis* (Ea). 3B, combination of *B. adolescentis*, *B. longum*, and *B. bifidum*. Dosages of all treatments: *Fn* = 1:1. *** *p* < 0.0001 vs. control.

**Figure 4 cells-12-01244-f004:**
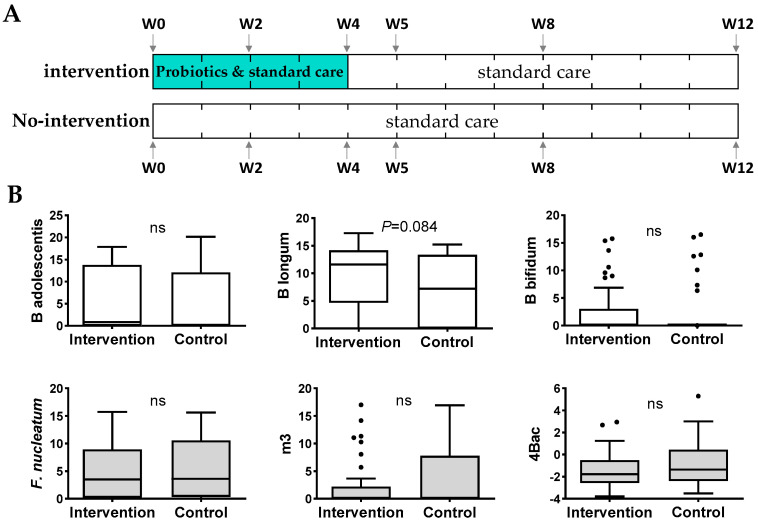
(**A**) Flow diagram of the pilot clinical study. The recruited patients in the intervention group took probiotic capsules daily from baseline (week 0, W0) to W4. Stools were collected from both groups of patients at W0, W2, W4, W5, W8, and W12. All patients in both intervention and no-intervention groups received the same treatment protocol for COVID-19 endorsed by the local health authority. (**B**) Baseline levels of *B. adolescentis*, *B. longum*, *B. bifidum,* and the CRC-related microbial markers showed no significant difference between the intervention and no-intervention groups. ns, not significant.

**Figure 5 cells-12-01244-f005:**
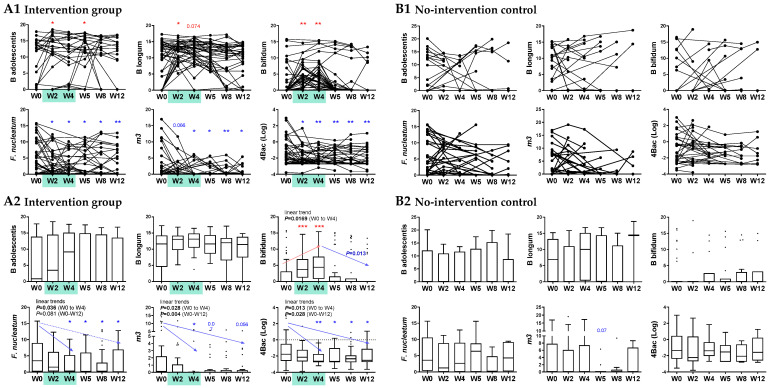
Intervention with the probiotic formula containing *B. adolescentis*, *B. longum,* and *B. bifidum* reduced microbial risk for CRC. (**A**) Changes in *B. adolescentis*, *B. longum,* and *B. bifidum* and the CRC-related microbial markers were detected in the intervention group by paired *t*-test vs. W0 (**A1**) and Mann–Whitney U test vs. W0 (**A2**). (**B**) No significant changes in *B. adolescentis*, *B. longum*, *B. bifidum*, and the CRC-related microbial markers were found in the no-intervention group by paired *t*-test vs. W0 (**B1**) or Mann–Whitney U test vs. W0 (**B2**). * *p* < 0.05, ** *p* < 0.001, *** *p* < 0.0001; red denotes increase and blue denotes decrease. Changes are not significant if not indicated.

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
