# Peer review of "A Probiotic Formula for Modulation of Colorectal Cancer Risk via Reducing CRC-Associated Bacteria"

_cells, 2023, doi:10.3390/cells12091244_

Round 1
Reviewer 1 Report
This study builds on previous data from the group, and provides an in depth analysis of the effect of a probiotic cocktail on the presence of CRC-associated bacteria including F. nucleatum. After demonstrating in vivo inhibitory effect of three Bifidobacteria species, an in vivo pilot was re-analysed to demonstrate the effect of these bacteria on intestinal microbiome.
The study was well performed and is of potential clinical relevance. There are a few issues that may be addressed to improve the study.
1.First, while I appreciate that PCR-based methods may have more relevance in clinical diagnostic practice in future, and while some individual bacteria have been shown to be associated with CRC, community structure may also be important, and may be used to infer the abundance of for instance SCFA producers etc. Metagenomic sequencing was performed, so perhaps functional analysis could also be performed?
2.The probiotics were supplemented with prebiotics, making it difficult to assess to what extent the pro and pre-biotics contribute to the effect on the intestinal microbiome. Perhaps endogenous levels of Bifidobacteria would already have been raised by the prebiotics? In addition, it is unclear whether active colonization takes place when measuring fecal levels of bifidobacteria, transient passage may account for the detected bacteria (also suggested by the fact that bacterial levels revert upon cessation of pro/prebiotic supplementation). Perhaps it would be more fair to speak of combination treatment than pro-biotic formulation, and discuss these issues in the discussion.
3. The study population is based on patients with COVID, which the authors mention in reference 21 already have gut dysbiosis. This makes it hard to extrapolate as to the effect of the pre/probiotics in the healthy/general population.
4. While the in vitro experiments prove causality at least to some extent, some of the (inverse) correlations between probiotic bacteria (figure 2) and pathogenic signatures, while significant, are quite weak, and one might argue how biologically relevant this will be for the course of CRC development. Stronger correlations may be present within the sequencing data with other bacteria.
5. the non-intervention group seems to have a lot more missing data, at least, there appear to be less lines in Figure 5? This would making it more difficult to show differences over time in this group perhaps.
Minor:
1. What was the reason for choosing The ratio of B. adolescentis to B. longum to B. bifidum is 4.6: 2.7: 2.7?
2. Please elaborate on the methods - how were copies of bacteria calculated for in vitro experiments (and why not primer-probe and deltaCT as for samples?).
3. in Fig 5A1: in the graphs with crc bacteria of intervention patients, shouldn’t week 2-4 also be indicated in blue for consistency?
4. In the discussion, the authors state ‘ here we showed the effects of our probiotic formula on reducing the microbial risk for CRC’. While nicely phrased, I would suggest that this may imply an anti-tumorigenic effect that has not been proven here. So I would suggest to rephrase to something like ‘ here we showed the effect of our probiotic formula on reducing levels of CRC-associated bacteria’.
Reviewer 2 Report
Lian et al. first found that three bifidobacteria (B. adolescentis, B. longum and B. bifidum) are significantly decreased in CRC patients. Second, they found their inhibitory effect on the growth of Fusobacterium nucleatum (Fn) in vitro system. Lastly, they developed a probiotic formula containing three bifidobacteria, and administration of this formula successfully decrease CRC-related bacteria, such as Fn and Lachnoclostridium sp. m3. This paper is focusing on a hot topic field and the reviewer believes this manuscript is appropriate for Cells after the revisions. To improve the manuscripts, the following points should be addressed before publication.
Major Comments:
1. The authors show the inhibitory effect of bifidobacterial on Fn growth in Figure3. However, Figure1 reveals that the number of these species decreases and F. nucleatum (Fm) is increased in CRC patients. Why does the proportion of Fn increase in the presence of these probiotics in the fecal samples from CRC patients? Is there any evidence that probiotics decrease at the early stages of CRC prior to the increase of Fn? In vivo (Fig1,2) and vitro (Fig3) data look paradoxical for the reviewers. They should discuss this point in the manuscript.
2. In Figure3, the authors show the inhibitory effect of bifidobacterial on Fn growth. How about this bifidobacterial growth in the presence of Fn? Does mixing with Fn show the beneficial effect of bifidobacterial growth? In addition, they only show the results of Fn growth. How about the growth of 4Bac which is also a CRC-related microbial marker and responds to the intervention more sharply than Fn in Fig4? They should address the 4Bac growth in vitro mixture system.
3. In Figure4, the baseline levels of three bifidobacterial, Fn, m3, and 4Bac are very different by individual. Are their age, sex, medical history, and internal medicine history similar to each other? If it is available, they should provide some background information in the manuscript.
Reviewer 3 Report
This is a good study showing the impact on probiotic species that are reduced in coloractal carcinoma (CRC) on the growth of CRC-associated bacteria (Fusobacterium).
* How Bifidobacterium sp-mediated suppression of F. nucleatum in in-vitro studies? Secretion of any inhibitory compounds?
* What would be the baseline levels of CRC-markers in the probiotic intervention and control groups in the pilot study, compared to the actual CRC patients?
* Fig 2 and 4B are difficult to read. The legends need to be clearly readable.
* I do not see any discussion on the possible differences in the strains of the probiotic species between the in vitro studies and the CRC patients.
Round 2
Reviewer 3 Report
The authors have responded to all of my comments.